# Impact of vaccination education in cardiac rehabilitation on attitudes and knowledge

**Andrea Rivera Solera[1,2], Marta Supervia[2,3,4], Jose R. Medina Inojosa[3], David Bedos Senon[2], Francisco Lopez-Jimenez[3], Sherry L. Grace** [ID][5,6] *

**1** Department of Physical Medicine and Rehabilitation, Alcala University, Madrid, Spain, **2** Department of Physical Medicine and Rehabilitation, Gregorio Marañón Health Research Institute, Gregorio Marañón General University Hospital, Madrid, Spain, **3** Division of Preventive Cardiology, Department of Cardiovascular Medicine, Mayo Clinic, Rochester, Minnesota, United States of America, **4** Faculty of Physical Activity and Sport Sciences, Universidad Politécnica de Madrid, Madrid, Spain, **5** Faculty of Health, York University, Toronto, Ontario, Canada, **6** KITE Toronto Rehabilitation Institute and Peter Munk Cardiac Centre, University Health Network, University of Toronto, Toronto, Ontario, Canada

* sgrace@yorku.ca

**Data Availability Statement:** Data are available with publication as supporting information file.

**Funding:** The authors received no specific funding for this work.

## Abstract

Clinical guidelines recommend influenza vaccination for cardiac patients, and COVID-19 vaccination is also beneficial given their increased risk. Patient education regarding vaccination was developed for cardiac rehabilitation (CR); impact on knowledge and attitudes were evaluated. A single-group pre-post design was applied at a Spanish CR program in early 2022. After baseline assessment, a nurse delivered the 40-minute group education. Knowledge and attitudes were re-assessed. Sixty-one (72%) of the 85 participants were vaccinated for influenza, and 40 (47%) for pneumococcus. Most participants perceived vaccines were important, and that the COVID-19 vaccine specifically was important, with three-quarters not influenced by vaccine myths/misinformation. The education intervention resulted in significant improvements in perceptions of the importance of vaccines (Hake's index 69%), understanding of myths (48%), knowledge of the different types of COVID vaccines (92%), and when they should be vaccinated. Vaccination rates are low despite their importance; while further research is needed, education in the CR setting could promote greater uptake.

## Introduction

Cardiovascular diseases (CVD) are a leading cause of mortality and morbidity globally [1]. Secondary prevention is achieved through heart-health behaviours, such as tobacco cessation, diet, physical inactivity, as well as medication adherence and influenza vaccination according to more recent clinical practice guidelines [2].

Cardiac rehabilitation (CR) is an effective chronic disease management model consisting of physical training sessions as well as education and counselling regarding these behavior changes [3]. Respiratory infections such influenza and pneumococcus–and more recently coronavirus disease (COVID-19)–are associated with poorer outcomes in CVD patients [4]. However, despite demonstrated efficacy and safety, not all CVD patients get vaccinated. This

**Competing interests:** The authors have declared that no competing interests exist.

is due to lack of awareness of the availability, Importance and/or safety of vaccination, lack of availability of vaccination in cardiac care centres, concern over side effects, among other factors [5]. Therefore, patient education regarding vaccination must be augmented in CR programs [6].

Accordingly, our CR program has augmented its' education, to raise awareness and educate cardiovascular patients about both influenza and COVID-19 vaccines. The aim of this study was to evaluate the effect of this education on vaccination knowledge and attitudes.

## Methods

A quasi-experimental study was conducted, with a single-group pre-post design. Ethics approval was obtained (RCVAC22) and written informed consent was secured from all willing participants.

The study was undertaken at the Gregorio Marañón General University Hospital CR program (February to April 2022) in Spain, where patients with guideline-indicated cardiac conditions are referred. A public system is used in Madrid so vaccines and CR are freely available. Participant inclusion criteria were: attending CR sessions at least weekly and attending the educational session on vaccination. Exclusion criteria were: cognitive impairment that prevented them from attending the educational intervention and/or filling out the surveys.

The questionnaire items were developed by the authors, and piloted in several patients. Participants were asked about sociodemographic characteristics, influenza and pneumococcal vaccination status and reasons for non-vaccination where applicable (open-ended), as well as their attitudes and knowledge towards influenza and COVID-19 vaccination (the latter were included in both pre and post surveys). Response options were on 5-point Likert scales or categorical.

The educational intervention was delivered by a nurse after the completion of the questionnaire in a group setting in accordance with other education sessions in the program. It consisted of a 40-minute face-to-face talk in Spanish covering: how vaccines work, benefits and side effects and vaccination schedules (available from corresponding author upon request). It was facilitated through a 21-slide Powerpoint presentation, followed by a question-and-answer period. After the session, they were re-administered questionnaire.

Descriptive statistics and paired t-tests were performed with IBM SPSS Statistics for Windows, Version 21.0. Armonk, NY. Valid percentages were reported in the case of missing responses. The learning gain was evaluated by calculating Hake's factor, a measure that compares the results of an initial and final test and allows us to obtain the degree of achievement of the educational intervention [7].

## Results

One hundred and seventeen patients were approached; 85 (72.6%) participants consented. Their characteristics are shown in Table 1.

Overall, 61 (72%) were vaccinated for influenza, and 40 (47%) for pneumococcus, with those over 65 years being more likely to be vaccinated with both (p < .05). Of those who had not been vaccinated for influenza (42%, n = 19), the main reasons were neglect or side effects. For those who had not been vaccinated for pneumococcus (53%, n = 30), the primary reason was lack of knowledge of the vaccine.

Table 2 displays vaccination knowledge and attitudes pre-education. Most perceived vaccines were important, and that the COVID-19 vaccine specifically was important, with three-quarters of participants impervious to vaccine myths/misinformation. mRNA vaccines were rated as safer.

**Table 1. Participant characteristics.**

| Characteristic | Mean ± SD / n (%) |
|---|---|
| Age (years) | 59±11.63 |
| Sex (% female) | 27 (31.7%) |
| Work Status | |
| Retired | 35 (43.2%) |
| Professional work | 23 (28.4%) |
| Other work | 21 (25.9%) |
| Highest Educational Attainment | |
| Primary | 14 (16.9%) |
| Secondary | 16 (19.3%) |
| Vocational program | 25 (30.1%) |
| University | 27 (32.5%) |

SD, standard deviation

Note: valid percentages reported due to some missing data.

When analyzing change to determine if the intervention had an effect, significant differences in the importance of vaccines were found (Table 2), with a learning gain (Hake's index) of 69%. In evaluating the myths about vaccines, we found significant changes, with a gain of 48% from pre- to post-intervention. Regarding knowledge about the functioning of the different COVID-19 vaccines, a learning gain of 92% was achieved. Regarding awareness of when they should be vaccinated for influenza and pneumococcus, a gain of 72% and 76% were achieved respectively.

**Table 2. Cardiac rehabilitation participant attitudes towards and knowledge regarding influenza and COVID-19 vaccination pre- and post-education intervention, N = 85.**

| Items | PRE | POST | P |
|---|---|---|---|
| **Importance of vaccines*** | 4.77 (± .68) | 4.93 (± .24) | .047 |
| **Attitudes (n, % yes)†** | | | |
| Vaccines contain substances that are hazardous to the body. | 8% (n = 66) | 5% (n = 66) | .209 |
| Vaccines cause many harmful side effects | 6% (n = 66) | 3% (n = 66) | .079 |
| Natural protection is better than vaccine-induced protection | 14% (n = 66) | 5% (n = 66) | .028 |
| Most people who get sick are those who have been vaccinated. | 5% (n = 66) | 3% (n = 66) | .329 |
| Vaccines cause autism | 0% (n = 66) | 0% (n = 66) | < .001 |
| None of the above are true | 77% (n = 66) | 88% (n = 66) | .044 |
| **Do you know how the different COVID-19 vaccines work? (n, % yes)** | 39% (n = 66) | 95% (n = 66) | < .001 |
| **How safe do you think the following COVID-19 vaccines are?*** | | | |
| Pfizer | 4.01 (± 1.14) | 4.5 (± .21) | .006 |
| Modern | 4.00 (± 1.00) | 4.45 (± .74) | .002 |
| AstraZeneca | 3.74 (± .92) | 3.85 (± 1.08) | .247 |
| Janssen | 3.65 (± .99) | 3.94 (± 1.09) | .018 |
| **Do you know how often to get a flu shot?** | 82% (n = 66) | 95% (n = 66) | .005 |
| **Do you know how often you should be vaccinated against pneumococcus?** | 41% (n = 66) | 86% (n = 66) | < .01 |

Note: mean and standard deviation or n and percentage shown.

*Scored on a scale from one to five, with higher scores indicating greater agreement.

†All statements are false, and thus "none of the above" is the correct response.

## Discussion

Studies have shown that patient education can lead to the development of positive attitudes, thereby increasing vaccine acceptance [8]. Moreover, provider recommendation has been shown to be one of the strongest predictors of vaccination [9]. This has been among the first studies to develop and preliminarily evaluate an educational intervention regarding vaccination in CR. While further research is needed, as well as a multi-pronged approach including improving ease of vaccine access, results suggest the education intervention significantly improved vaccination knowledge and attitudes in the short-term, with change in almost all items assessed.

Social media has promulgated vaccine hesitancy in recent years, spreading negative and alarming messages [10]. The anti-vaccine myths that have appeared have contributed to the misinformation of patients, such that a quarter of patients believed some of the myths before the intervention. However, following exposure to the education, almost 90% of patients did not believe vaccination misinformation.

Caution is warranted when interpreting these results. The study was conducted on a convenience sample at a single centre, and therefore generalizability is likely limited. The non-randomized and non-controlled design precludes causal conclusions, such that future study with a more rigorous multi-centre design is needed. In future research, refinement of the educational intervention based on patient input should be undertaken (considering also length of session and volume of information), COVID-19 vaccination status should be assessed, and impact on vaccination behavior investigated.

In conclusion, despite that healthcare providers are to recommend influenza vaccination to CVD patients, vaccination rates are still poor. Given the importance now of COVID-19 vaccination as well as for influenza, a strategic approach is needed to improve vaccination rates, including better education [11]. Through this study, an educational intervention was shown to be effective in increasing vaccination knowledge and attitudes in CR patients.

## Supporting information

**S1 Data.**
(XLSX)

## Author Contributions

**Conceptualization:** Jose R. Medina Inojosa, Francisco Lopez-Jimenez.

**Data curation:** Andrea Rivera Solera, David Bedos Senon.

**Formal analysis:** Andrea Rivera Solera.

**Investigation:** Andrea Rivera Solera, David Bedos Senon.

**Methodology:** Francisco Lopez-Jimenez.

**Project administration:** Marta Supervia, Jose R. Medina Inojosa, Sherry L. Grace.

**Resources:** Marta Supervia.

**Supervision:** Marta Supervia.

**Validation:** Francisco Lopez-Jimenez.

**Writing – original draft:** Andrea Rivera Solera, Marta Supervia.

**Writing – review & editing:** Sherry L. Grace.

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
