## [Editor Report · Decision Letter 0]

3 Nov 2023

PGPH-D-23-02047

Impact of Vaccination Education in Cardiac Rehabilitation on Attitudes and Knowledge

Dear Dr. Grace,

Thank you for submitting your manuscript to PLOS Global Public Health.

Before I send this out to peer review, I have several suggestions.

1 - could you better explain Hake's gain and/or reference some resources? I'm not familiar with it, and after searching around, I do think it could be applicable here. Yet, I think the average readership for this journal would not quite understand it, so providing that info could be useful.

2 - you have a lot of outcomes, I believe around 12 in Table 1. Ideally you can do some sort of correction for multiple p-values. Alternatively, you could pre-specify in an aim what is your main outcome in a confirmatory analysis, and then put everything else in a separate table as an explanatory analysis (with the exploratory analyses correcting for multiple p-values.

---

## [Decision Letter · Decision Letter 1]

22 Jan 2024

PGPH-D-23-02047R1

Impact of Vaccination Education in Cardiac Rehabilitation on Attitudes and Knowledge

Dear Dr. Grace,

Thank you for submitting your manuscript to PLOS Global Public Health. After careful consideration, we feel that it has merit but does not fully meet PLOS Global Public Health’s publication criteria as it currently stands. Therefore, we invite you to submit a revised version of the manuscript that addresses the points raised during the review process.

We look forward to receiving your revised manuscript.

Kind regards,

Abram L. Wagner, PhD, MPH

Academic Editor

Journal Requirements:

Additional Editor Comments (if provided):

In the introduction could you mention lack of access to vaccination in cardiac facilities? Presumably one barrier is that as individuals go to rehab, there is not a convenient place to access vaccination at those locations. (Feel free to disagree)

Can you confirm details about the intervention. It was forty (40) minutes long? Was that just about vaccines? Or were there other components to that 40 minute talk? If it really was a 40 minute talk on vaccines, possibly worth mentioning a limitation being the substantial burden of time this intervention requires.

You could mention in the discussion that this research is in line with vaccine studies in general which have shown provider recommendations to be one of the strongest predictors of vaccination (many such references exist - but possibly you could find a review related to vaccination of individuals with chronic disease)

Reviewers' comments:

Reviewer's Responses to Questions

**Comments to the Author**

1. If the authors have adequately addressed your comments raised in a previous round of review and you feel that this manuscript is now acceptable for publication, you may indicate that here to bypass the “Comments to the Author” section, enter your conflict of interest statement in the “Confidential to Editor” section, and submit your "Accept" recommendation.

Reviewer #1: All comments have been addressed

Reviewer #2: (No Response)

2. Does this manuscript meet PLOS Global Public Health’s publication criteria? Is the manuscript technically sound, and do the data support the conclusions? The manuscript must describe methodologically and ethically rigorous research with conclusions that are appropriately drawn based on the data presented.

Reviewer #1: Yes

Reviewer #2: No

3. Has the statistical analysis been performed appropriately and rigorously?

Reviewer #1: Yes

Reviewer #2: No

4. Have the authors made all data underlying the findings in their manuscript fully available (please refer to the Data Availability Statement at the start of the manuscript PDF file)?

Reviewer #1: Yes

Reviewer #2: No

5. Is the manuscript presented in an intelligible fashion and written in standard English?

Reviewer #1: Yes

Reviewer #2: No

6. Review Comments to the Author

Reviewer #1: Congratulations to the authors of this study. It is a small ''exploratory'' study with clear limitations requiring further research with robust design with bigger sample size to measure especially the impact of the educational interventions for cases and controls.

Reviewer #2: Introduction

Line 49 - "Respiratory infections such influenza and pneumococcus -- and more recently coronavirus disease (COVID-19) – are associated with poorer outcomes in CVD patients." Can the authors provide references for the statement?

Line 55 - "Accordingly, our CR program has augmented our education, to raise awareness and educate cardiovascular patients about both influenza and COVID-19 vaccines. The aim of this study was evaluate the education efficacy in changing vaccination knowledge and attitudes." The sentences need to be improved, please provide more details for the objectives of the study.

Methods

Line 68 "The questionnaire items were developed by the authors, and piloted in several patients." Please provide references if other standardised questionnaires have been referred to and served as a source for the content.

Results

Line 85 "Eighty-five participants consented" could the authors also provide the total number of CR patients approached for the study? It will provide the reader the consent rate.

Please provide a Table of Basic Demographic of the participants.

Please provide a copy of the questionnaire.

Please provide a statement on data availability. For the attached Excel file, what are in the columns p4 and p5? Are they patients identifiers? Dat should be de-identified for information shared in public domains.

7. PLOS authors have the option to publish the peer review history of their article (what does this mean?). If published, this will include your full peer review and any attached files.

**Do you want your identity to be public for this peer review?** For information about this choice, including consent withdrawal, please see our Privacy Policy.

Reviewer #1: No

Reviewer #2: **Yes: **Sok King Ong

---

## [Editor Report · Decision Letter 2]

16 Feb 2024

Impact of Vaccination Education in Cardiac Rehabilitation on Attitudes and Knowledge

PGPH-D-23-02047R2

Dear Dr. Grace,

We are pleased to inform you that your manuscript 'Impact of Vaccination Education in Cardiac Rehabilitation on Attitudes and Knowledge' has been provisionally accepted for publication in PLOS Global Public Health.

Best regards,

Abram L. Wagner, PhD, MPH

Academic Editor